# Polymeric Nanocarriers: A Transformation in Doxorubicin Therapies

**DOI:** 10.3390/ma14092135

**Published:** 2021-04-22

**Authors:** Kamila Butowska, Anna Woziwodzka, Agnieszka Borowik, Jacek Piosik

**Affiliations:** 1Laboratory of Biophysics, Intercollegiate Faculty of Biotechnology, University of Gdansk and Medical University of Gdansk, Abrahama 58, 80-307 Gdańsk, Poland; kamila.butowska@phdstud.ug.edu.pl (K.B.); anna.woziwodzka@ug.edu.pl (A.W.); agnieszka.borowik@phdstud.ug.edu.pl (A.B.); 2Aging and Metabolism Research Program, Oklahoma Medical Research Foundation (OMRF), Oklahoma City, OK 73104, USA

**Keywords:** doxorubicin, drug delivery, polymers, targeted therapy, anticancer treatment, controlled release

## Abstract

Doxorubicin, a member of the anthracycline family, is a common anticancer agent often used as a first line treatment for the wide spectrum of cancers. Doxorubicin-based chemotherapy, although effective, is associated with serious side effects, such as irreversible cardiotoxicity or nephrotoxicity. Those often life-threatening adverse risks, responsible for the elongation of the patients’ recuperation period and increasing medical expenses, have prompted the need for creating novel and safer drug delivery systems. Among many proposed concepts, polymeric nanocarriers are shown to be a promising approach, allowing for controlled and selective drug delivery, simultaneously enhancing its activity towards cancerous cells and reducing toxic effects on healthy tissues. This article is a chronological examination of the history of the work progress on polymeric nanostructures, designed as efficient doxorubicin nanocarriers, with the emphasis on the main achievements of 2010–2020. Numerous publications have been reviewed to provide an essential summation of the nanopolymer types and their essential properties, mechanisms towards efficient drug delivery, as well as active targeting stimuli-responsive strategies that are currently utilized in the doxorubicin transportation field.

## 1. Introduction

### 1.1. Doxorubicin and Other Anthracyclines

Anthracyclines, including doxorubicin (DOX), daunorubicin, and epirubicin, are among the most active antitumor compounds with the widest spectrum of activity in human cancers such as carcinomas, sarcomas, and hematological malignancies. They are widely used (alone or in combination with other cytotoxic agents) in clinical practice for the treatment of lung, breast, ovarian, and urinary bladder cancers, as well as multiple myeloma, soft tissue sarcoma, osteosarcoma, leukemias, and Hodgkin’s lymphoma. DOX was initially obtained from *Streptomyces peucetius* actinobacteria isolated from a soil sample, identified, and developed in the 1960s [1,2]. Although DOX was granted marketing authorization nearly five decades ago, it is present on the current World Health Organization Model List of Essential Medicines, listing the most efficient, safe, and cost–effective medicines needed in the healthcare system [3].

Apart from its high efficacy in monotherapy (especially in treatment of metastatic breast cancer), several combination therapies including DOX were also developed. The combination of DOX with cyclophosphamide, vincristine, and prednisone is used for treatment of diffuse large cell non-Hodgkin’s lymphomas [4]. The combination of DOX with bleomycin, vincristine, and dacarbazine is beneficial and well tolerated in patients with Hodgkin’s lymphoma [4]. Several combination regimens consisting of DOX are used for treatment of breast cancer (DOX with cyclophosphamide and/or taxotere, DOX with cyclophosphamide and fluorouracil).

Structurally, DOX is a glycoside of anthracyclinone. It contains an anthraquinone chromophore placed within a planar aromatic system of four cycles, bound by a glycosidic bond to daunosamine (Figure 1). Anthraquinone groups can participate in redox reactions, contributing to the generation of reactive chemical species, which might be associated with anthracycline cardiotoxicity [5].

### 1.2. Doxorubicin Mechanism of Action

To date, several distinct mechanisms of DOX action are discussed (Figure 2). The first and primary one includes the interaction of DOX with mammalian topoisomerase II, stabilization of enzyme–DNA complex, and resulting inhibition of single- and double-strand DNA breaks re-ligation during the DNA replication process [6]. This leads to irreversible DNA damage and cell death. Importantly, this mode of DOX action is specific for proliferating (e.g., cancer) cells which, mitotically–active, are predominantly affected by topoisomerase II–induced DNA breaks [7]. Such a mechanism of action was confirmed in in vitro studies on cell lines with mutated or downregulated topoisomerase II, in which resistance to DOX was reported [8,9,10]. Intercalation of DOX into DNA double–helix is well–evidenced and widely accepted, and 5′TCA was reported as a consensus sequence for the highest DOX affinity [11]. Nevertheless, the actual role of DOX intercalation to DNA in topoisomerase II–mediated DNA damage remains unknown. Topoisomerase II–related DNA breaks are reported at DOX concentrations which fall below the DOX-DNA association constant, along with the fact that selected anthracycline analogs do not intercalate into DNA but still exert cytotoxic activity, might suggest that DOX intercalation to DNA is not essential for its interference with topoisomerase II [12,13].

Intercalation of DOX into DNA, although possibly not involved in targeting topoisomerase II, has an impact on several vital intracellular processes. It can affect the activity of enzymes involved in DNA replication and transcription, such as helicases, DNA, or RNA polymerases [14,15]. Topological DNA changes following DOX intercalation were also reported to be associated with increased nucleosome turnover around promoters, which affected levels of gene expression [16]. DOX-related removal of nucleosomes at open chromatin regions, which alters epigenetic regulation of transcription and contributes to reduced DNA repair of DOX–induced double–strand breaks, was recently reported [17].

Apart from its well–established topoisomerase II–mediated cytotoxicity, DOX, while undergoing intracellular oxidation and reduction cycles, leads to the generation of reactive oxygen species. This exposes nuclear and mitochondrial DNA to oxidative stress and can exert additional cytotoxic effects [18,19]. Indeed, oxidized DNA bases are detected in the blood and urine of patients treated with DOX [20,21]. Additionally, DOX was shown to form covalent adducts with DNA, which can induce apoptosis, further contributing to the overall cytotoxic activity of the drug [22,23]. All in all, at DOX concentrations reflecting peak plasma concentration during treatment, targeting topoisomerase II seems to be the primary mechanism of antitumor action, whereas, at higher drug concentrations, the toxicity of free radicals and DNA cross–linking may become relevant [24].

### 1.3. Limitations of DOX Therapy

Two important limitations associated with antitumor therapy with DOX are recognized: development of drug resistance and treatment toxicity, associated with the occurrence of serious adverse effects. The former include enhanced drug efflux (specific for anthracyclines and through multidrug resistance transporters), altered topoisomerase II activity, and enhanced antioxidant defense [25]. Cardiac toxicity, both acute and chronic, represents the major complication associated with DOX treatment and constitutes the main reason for dose-limited drug administration [26]. Acute cardiotoxic effects such as arrhythmias, hypotension, and electrocardiographic alterations are transient and disappear at treatment cessation. Chronic cardiotoxicity is dose-dependent; more than a quarter of patients receiving DOX with a cumulative dose of 550 mg/m^2^ would develop congestive heart failure [27]. The mechanism responsible for DOX heart toxicity is not fully understood, but oxidative stress disrupting major mitochondrial functions is considered the most presumable.

DOX induces myelosuppression, mainly in the form of leukopenia (principally granulocytopenia), neutropenia, or thrombocytopenia, with up to 80% of patients treated with conventional doses of DOX being affected [4,28]. The severity of myelotoxicity is dose-dependent; therefore, it represents the major dose-limiting side effect of anthracycline therapy.

Besides the heart and bone marrow, toxic effects of DOX are also observed in the liver, kidney, and brain [26]. Other side effects of DOX include nausea and vomiting, stomatitis, mucositis, alopecia, and neurologic disturbances (dizziness, hallucinations) [4,29]. Severe vesicant reactions might also occur upon extravasation of DOX which can lead to severe local tissue necrosis and reduced mobility in the adjacent joints.

Cancer survivors in childhood have more than a two–fold increased risk of acute leukemia and solid tumors at the age of 40, and the history of DOX treatment has a well–established association with the development of secondary cancer [30].

For decades, significant effort has been made to develop new anthracycline derivatives that would markedly reduce DOX toxic effects and at least maintain its antitumor activity [31,32]. Although a few of them (e.g., epirubicin, idarubicin, valrubicin) were granted marketing authorization, no evident or clinically relevant benefit in terms of enhanced effectiveness and/or improved safety profile has been achieved so far. More recently, heteroarene–fused anthracenediones, a combination of anthraquinone and polyphenolic structures, and bis–intercalating agents, have been described as novel promising approaches [33,34,35].

The rapid development of novel drug delivery systems (DDSs), which are aimed at directing the drug specifically to neoplastic cells, provides promising tools to minimize DOX systemic toxicity. Such an approach, while maintaining DOX satisfactory profile of antitumor activity, would allow the delivery of higher doses of the drug directly to the cancer cells. Here, we review recent advances on new platforms of targeted DOX delivery.

## 2. Evolution of Drug Delivery Systems

### 2.1. From Macro- to Nanoscale

The history of DDSs stretches back to 1960 when Folkman discovered a constant rate drug delivery implant for prolonged therapy used a silicone rubber tube (Silactic*^®^*) loaded with the drug [36,37]. The seminal work of Folkman et al. was an inspiration for scientists who focused on new concepts of zero–order-controlled drug delivery in the macroscale using various types of polymers in a wide field of medicine. In the following years, Ocusert*^®^* containing an anti–glaucoma drug, Progestesert*^®^* releasing progesterone in the uterine cavity, or Implanon*^®^* as sub-dermal devices were developed [38]. In 1976, Folkman and Langer reported a pioneering work showing that proteins and other macromolecules (large molecular weight drugs) could undergo sustained release from non-inflammatory polymers [39]. On the turn of the 1980s and 1990s, other strategies of zero–order DDSs with controlled diffusions such as skin patches and osmotic capsules were investigated. Since the first demonstration of low and large molecular weight drug delivery matrices, DDSs have evolved from zero–order macroscale systems to biodegradable microscopic polymers, using poly(glycolic acid) (PGA), poly(lactic-co-glycolic acid) (PLGA), or copolymers of PGA–PLGA [40]. Then, various approaches were adopted to deliver drugs by rationally designed polymers enter the nano-sized era and showed significant therapeutic potential [41,42]. Indeed, polymeric systems such as polymer–drug conjugates, block copolymers, and polymer–protein conjugates, also lipid and inorganic nanoparticles or multicomponent systems, were widely utilized in combination with therapies [43,44,45,46]. During the last decade, there has been significant progress in the development of high–performance DDSs. They became increasingly complex, and it became possible to control their chemical and physical properties. Since many aspects of these topics were thoroughly described in previous reviews, we focused on the latest trends in the doxorubicin delivery systems combined with increasingly innovative systems [47,48].

### 2.2. Bringing New Life to Carriers

All above mentioned polymeric subclasses used specific polymers with exceptional properties to develop sophisticated and biodegradable DDSs in nanosize ranging from 1 to 100 nm [49,50]. Polymeric-based nanoparticles (PNPs), based on natural and synthetic polymers, have various physicochemical properties, and different architectures and sizes, which allow them to carry drugs to the target [51]. Therefore, the choice of the polymer, drug loading, and shape are crucial for the design of PNPs in a controlled manner to achieve the desired DDSs (Figure 3). Additionally, PNPs show significant solubility and stability, higher targeting specificity, and exhibit controlled drug release by carrier degradation, diffusion through carrier matrix, or dissociation mechanisms e.g., photo–dissociation [52]. From the biological standpoint, polymeric nanocarriers showed a longer half–life in pharmacokinetic studies and have an enhanced permeability and retention effect which allows them to accumulate in cancer tumors rather than in healthy tissues [53]. With this fact in mind, many natural and synthetic polymers, as well as pseudosynthetic ones, attract attention in medicine, as antineoplastic or antimicrobial drug carriers (Figure 4). To note, natural polymers are more biocompatible than synthetic, nevertheless, some natural polymers are highly immunogenic [54]. On the other hand, synthetic polymers are less biodegradable than the natural ones, but this may be altered through structural modifications. Hence, current efforts focused on synthetic polymers to control the monomer class and its ratio, as well as molecular weight and crosslinking of the polymer. Modern polymer chemistry takes advantage of different structures, from a linear block and gradient copolymers to increasingly intricate polymers, including stars, combs, and brushes, to dendronized and (hyper)branched polymers [55,56,57]. This demanded many polymerization methods to be employed for polymers to be formed in a piece–by–piece fashion. The most effective and widely used methods are controlled radical methods, such as reversible additional fragmentation chain transfer (RAFT), and atom transfer radical polymerization (ATRP), which were reported as more effective than conventional polymerization techniques [58,59]. Considering the wide spectrum of polymers and efficient polymerization methods, numerous potential DDSs appeared to offer many advantages including self–assembly, biocompatibility, and high loading capacity.

After years of research, Doxil^®^—pegylated liposomal DOX delivery systems—was approved by Food and Drug Administration (FDA) in 1995. Additionally, Myocet^®^ (non-pegylated liposomal DOX) in 2000 has received Fast Track Designation from FDA for the treatment of HER2 positive breast cancer and has been approved in Europe and Canada). Despite the well–known and approved DOX delivery systems, efforts continued to develop more efficient and safe carriers [60,61].

The first natural polymer–DOX conjugate, called AD–70, which entered clinical trials in 1993, employed polymer derivatives of the oxidized dextran (DX) coupled with DOX (DX–DOX) via Schiff base [62]. AD–70 conjugate was highly selective for DOX delivery in an animal model; unfortunately, in a Phase I clinical study, substantial toxicity was observed leading to thrombocytopenia and hepatotoxicity in the patients.

In the following years, Mitra et al. encapsulated DX–DOX conjugate into chitosan (CS) nanoparticles using reverse microemulsion [63]. This resulted in faster regression of tumor volume from 514 ± 6 mm^3^ in the middle of treatment to 170 ± 7.3 mm^3^ at day 90. Throughout 90 days of the study, Balb/C mice treated with DX–DOX encapsulated into CS showed almost 50% survival rate, while mice treated with DX–DOX demonstrated only a 20% survival rate. Furthermore, Janes et al. described a similar conception that included encapsulation of DOX into CS nanoparticles (with encapsulation efficiency ~20%) through the charge repulsion between the polymer and the drug. Encapsulation of the drug in CS was possible via the interaction of a DOX amino group with incorporated DX sulfate [64]. Another strategy for designing DDSs, reported in 2010 by Qi et al., used a simple protocol to develop biocompatible bovine serum albumin (BSA)–DX–CS nanoparticles by heating, with DOX loaded into nanoparticles by diffusion following pH change from 5.4 to 7.4 [65]. Hepatoma H22 tumor–bearing mice treated with 12.0 mg/kg of DOX nanoparticles had prolonged life from 10.3 to 14.8 days, but tumor growth was reduced less effectively compared with free DOX. Similarly, in the study by Du et al., BSA was used to synthesize a water–soluble DOX delivery system with higher tumor selectivity achieved by linking to folic acid (FA), which binds to folate receptors over–expressed on the surface of mammary human cancer cells [66]. With the continuing desire to increase the DOX loading and entrapping capacity into a carrier, Maspoch’s group prepared coordination polymer particles generated by connecting Zn^2+^ metal ions through 1,4–bis(imidazol–1–ylmethyl)benzene organic ligands (bix) via coordination polymerization followed by fast precipitation [67]. DOX entrapped into Zn(bix) showed ~80% of drug released in PBS pH = 7.4 at 37 °C within 8 h, suggesting gradual erosion of Zn(bix) in time. DOX/Zn(bix) diminished human promyelocytic leukemia HL60 cells viability to 25% at higher concentrations ~10 µM with IC_50_ of 5.2 μM. Against the HeLa cell line, Mrówczyński et al. developed polydopamine coated Fe_3_O_4_ nanoparticles through a coprecipitation method and oxidative polymerization of dopamine loaded with DOX [68]. The maximum of DOX release was achieved after 24 h. The cellular study against HeLa cells showed that after three days of incubation, cell viability dramatically decreased to 6% at a concentration of 100 μg/mL.

Currently, the rise of nanotechnology and polymer science provided many novel DDSs for efficient anticancer therapy by rational design, and allows one to study the behavior of nanoparticles on the cellular level. The theoretical and experimental findings shown in 2020 by Zhang et al. demonstrated the criteria for the preparation of new fluorinated polymers for DDSs, denoted poly(oligo(ethylene glycol) methyl ether acrylate)_m_–perfluoropolyether (poly(OEGA)–PFPE; where m = 5, 10, and 20) [69]. Block copolymers containing OEGA and PFPE with different fluorine contents (28.7 weight percentage (wt.%) [m = 5, named P5], 17.0 wt.% [m = 10, named P10], and 9.8 wt.% [m = 20, named P20] were prepared through RAFT and conjugated with DOX via a hydrazine bond. Molecular dynamics (MD) simulations were consistent with experimental results and showed single–chain folded conformation of DOX–conjugated P20, whereas DOX–conjugated P5 and DOX–conjugated P10 formed micelle-like assemblies. Moreover, MD results, performed with NAMD code, investigating interactions between DOX–conjugated poly(OEGA)_m_–PFPE with a cell membrane, highlighted faster diffusion across the membrane of DOX–conjugated P20 than P5 and P10 because of its small hydrophobic core (PFPE). Furthermore, DOX–conjugated P20 showed higher cellular uptake and therapeutic efficacy toward breast cancer cell line MCF–7 than the P5 and P10.

## 3. Stimuli-Responsive Drug Delivery Systems

### 3.1. Choose Your Target

Further studies showed that polymers can be combined with inorganic nanoparticles and small molecules to create stimuli-responsive or targeted DDSs (Figure 5) [70,71,72,73]. The targeting of DDSs focuses on both active targeting and improving the efficacy by stimuli–responsive approaches. For example, monoclonal antibodies (mAbs) are becoming increasingly popular, i.e., trastuzumab, cetuximab, or bevacizumab, and, apart from their intrinsic anticancer activity, are proposed to be used for selective delivery of antineoplastic drugs to tumors [74,75,76,77]. Additionally, to achieve active targeting, a large number of ligands have been employed, including polysaccharides and peptides (i.e., hyaluronic acid and RGD peptide), as well as small molecules like folate or anisamidephenylbornic acid [78,79]. Furthermore, overexpression of enzymes, i.e., proteases, is another approach that can be used to design responsive DDSs [80,81]. In the enzyme–sensitive DDSs, the peptide side chain is designed as a specific substrate of a target enzyme that could directly release the drug from a carrier. Other promising strategies include chemical stimuli–responsive DDSs that can release the drug from a carrier by pH changes and using acid–labile or redox–responsive chemical bonds [82,83]. Among the common physical stimuli, thermo/magnetic-responsiveness and light/ultrasound-triggered stimulus are the most frequently used [84,85,86]. For all these features, targeting strategies of DDSs present an exciting approach for anticancer treatment.

### 3.2. Drug Delivery Systems Responsive to Physical and Chemical Stimuli

Cancers are known to acidify their environment by dysregulation of pH dynamics. During neoplastic progression, the extracellular pHi of cancer decreases to 6.8 compared with normal cells (7.4), whereas intracellular pH increases to 7.3–7.6 (vs. 7.05–7.2 in normal cells) [87]. Moreover, membrane-bound organelles such as endosomes and liposomes, involved in the endocytic pathway, which is a specific mechanism for some DDSs to enter cells, exhibit remarkably lower pH, approximately 5–6 and 4.5–5, respectively [88,89].

These properties provide a rationale to design a prodrug–based carrier with the time–dependent drug release behavior in the acidic environment of cancer, reported by Zhang et al. [90]. Designed prodrug (DOXDT) consisted of dextran–poly(oligoethylene glycol) methyl ether methacrylate–*co*–methyl glycol methacrylate copolymer prepared by one–step ATRP and conjugated with DOX, forming stable micelles. DOXDT showed pronounced tumor permeability and cytotoxicity. In vivo studies showed that Balb/C mice bearing 4T1 tumors treated with DOXDT (DOX dosage, 5 mg/kg) suppressed the tumor with an 85.5% inhibition rate, and was far more effective than free DOX. Importantly, DOXDT presented a good safety profile toward major organs, including the heart, liver, spleen, lung, and kidney, and minimal systemic toxicity.

Investigations carried out by She et al. showed that dendronized heparin–DOX could be also useful for pH-stimuli delivery of antineoplastic drugs [91]. The dendron conjugated through the hydrazine bond to DOX was attached to azido–heparin via click reaction, resulting in a self–assembled nanocarrier. DOX release from nanocarrier was faster and higher at pH 5.0 (80% of drug release after 56 h) than at the physiological pH of 7.4. In addition, both in vitro and in vivo studies showed high 4T1 breast tumor inhibition and no significant toxicity toward healthy organs.

Due to the cancer acidic environment, PLGA–coated stabilized (Mn, Zn) ferrite nanoparticles loaded with DOX (DOX–PLGA@CS@Mn_0.9_Zn_0.1_Fe_2_O_4_) were designed for pH–triggered DOX release [92]. The pH change from physiological to acidic resulted in a significant increase in the DOX release rate (34.26% for physiological pH vs. 57.18% for acidic pH). DOX–PLGA@CS@Mn_0.9_Zn_0.1_Fe_2_O_4_ was less cytotoxic (from 3 to 125 µg/mL) against HeLa cells compared with free DOX, while at higher concentrations (250 µg/mL) its cytotoxicity was similar to that of DOX.

In an effort to further improve DOX release performance of the DDSs, dual or multi-stimuli responsive DDSs were recently developed [93]. Novel DOX–CuCo_2_S_4_@PIL nanocarrier, proposed by Fan et al. to be effective in anticancer treatment, responds to both pH– and thermo–stimuli. The primary prepared Cu Co_2_S_4_ nanoparticles were subsequently modified with the poly(tetrabutyl phosphonium styrenesulfonate) (PIL), then the DOX was loaded onto PIL. CuCo_2_S_4_ utilized the near–infrared (NIR) irradiation to convert light energy into heat to destabilize the PIL and promote drug release. The DOX release of DOX–Cu Co_2_S_4_@PIL at 45 °C and pH 5.0 reached 90.5% compared with 79.5% at 37 °C. At pH 7.4, the release ratio was only 21.8% (37 °C) and 20.5% (45 °C). The in vitro analysis against MCF–7 cells showed the biocompatibility of CuCo_2_S_4_@PIL carrier even at high concentration. The cytotoxic effects were much higher when the cells were treated with DOX–Co_2_S_4_@PIL in the presence of NIR laser irradiation at 808 nm than without such irradiation. The in vivo effects of DOX–Co_2_S_4_@PIL on the breast tumor-carrying mice were assessed 16 days following the treatment. DOX–Co_2_S_4_@PIL with exposure to NIR laser irradiation at 808 nm resulting in improved tumor inhibition, whereas DOX–Co_2_S_4_@PIL without NIR laser irradiation displayed tumor inhibition the same as free DOX.

Several reports described stimuli–responsive three–dimensional hydrogels as smart DDSs. Xiong et al. prepared the pH– and temperature–responsive nanogels consisting of poly(N-isopropylacrylamide–*co*–acrylic acid) and DOX (DOX–PNA) as promising DDSs against human liver carcinoma cells HepG2 [94]. Under hyperthermia of 43 °C at pH 6.8, the cytotoxicity of DOX–PNA increased by approximately 43% when compared with the equivalent dose of DOX–PNA at 37 °C and pH 7.4.

Omidi et al. developed pH–responsive DOX–loaded hydrogel composed of CS, aminated–graphene, and amino–functionalized cellulose nanowhisker cross–linked by dialdehyde (DOX–CGW) [95]. Field Emission Scanning Electron Microscopes images showed a randomly porous structure with DOX accumulated on the surface of CGW, which remained stable at PBS buffer (pH 7.4) after 6 h, contrarily to distilled water. The significant DOX release rate (63%) from CGW was observed at pH 5.4, whereas approximately 35% of the drug was released at pH 7.4. Ultimately, subcutaneous injection at the back of the rat led to in vivo hydrogel formation 2 min after the injection. This provided a basis for further engineering of CGW as injectable DDSs.

A tremendous amount of work has been done to predict the drug release behaviors of stimuli–responsive hydrogels with artificial intelligence–based techniques such as Artificial Natural Networks (ANNs), Support Vector Machine (SVM), and its adaptation—Support Vector Regression (SVR) [96,97]. Boztepe et al. used these methods to predict the DOX release behavior of interpenetrating polymer network (IPN) hydrogel. IPN hydrogel based on poly(N–isopropyl acrylamide–*co*–acrylic acid and poly(ethylene glycol) was synthesized by free radical solution polymerization and loaded with DOX (64.81 mg DOX/g polymer) [98]. The DOX release rate was much more rapid at acidic pH and at a temperature above the lower critical solution temperature. The most efficient DOX release from IPN hydrogel was observed at pH 4 and 45 °C (88%), whereas at pH 7 at the same temperature DOX release was two times lower (~40%). Further mathematical ANN studies showed agreement between prediction and observations (i.e., experimental DOX release kinetic data), which proves its usefulness as a tool for the rational design and modeling of DDS-like hydrogels.

Meanwhile, Zhang et al. reported the efficacy of dual–sensitive (pH and redox) nanogels (DSNGs) against triple–negative breast cancer by hydrogel–assisted delivery [99]. Hydrogel composed of oxidized DX was crosslinked by imine bonds with 25% G5–PAMAM dendrimer that degraded under hydrolytic conditions [100]. Furthermore, DSNGs based on oxidized DX were crosslinked with cystamine, introducing a redox–sensitive disulfide bond cleaved in the presence of glutathione–reductant in cancer cells. Additionally, DOX was conjugated by a pH–sensitive imine bond to DX. DSNGs were released from degraded hydrogel, followed by a rapid release of DOX in cancer cells. Cell viability toward MDA MB 231 and 3T3 cell lines treated by DSNGs showed significantly higher toxicity in the presence of glutathione (IC_50_ values equal to 114 and 2338 nM, respectively), whereas in vivo studies showed tumor value reduction in the first 24 h post–injection, but slow tumor growth up was accelerated at 72 h, which may limit DSNGs applications.

Recently, Biswas et al. developed PEG functionalized guanosine–quadruplex–based hydrogel (G4PEG) to produce stimuli–responsive DDSs with zero–order DOX release [101]. It is well known that 1,2–cis–diol of guanosine forms dynamic boronate ester bonds with 2-formylphenylboronic acid (FPBA). Moreover, FPBA simultaneously forms dynamic imine bonds with primary amines such as 4-arm PEG–NH_2_. Thus, the working mechanism of G4PEG is believed to depend on iminoboronate bonds, which are unstable at lower pH, resulting in sustained DOX release. DOX release rates obtained for acidic and physiological pH were 7.4 × 10^−5^ and 2.25 × 10^−5^ mmol/sec, respectively. The cell viability MTT assay using MCF–7 cell line showed weak, concentration–dependent cytotoxic effects of G4PEG with an IC_50_ value of approximately 2.27 mM. For DOX-loaded G4PEG, the IC_50_ value was lower (1.3 mM).

### 3.3. Mitochondrial-Targeting Drug Delivery Systems

Interestingly, despite many unique characteristics of cancer cells, like low extracellular pH and hypoxia, their hyperpolarized mitochondria opened new directions to targeted drug delivery [102]. Many reports demonstrated potential applications of modified polymers to locate drugs inside the mitochondria [103,104].

In 2019, Tan et al. presented micelles for DOX delivering, using glycolipid polymer chitosan-stearic acid (CSOSA), which was modified by lipophilic (4-carboxybutyl)triphenylphosphonium bromide (CTPP) cations, to form mitochondria-targeted DDSs (C-P-CSOSA/DOX) [105]. The relatively small C-P-CSOSA/DOX particles, with a size around 100 nm, showed higher cellular uptake in human breast adenocarcinoma cells (MCF-7 cell line) than in human normal liver cells (L02 cell line). Importantly, C-P-CSOSA/DOX demonstrated efficient colocalization into mitochondria in vitro and in vivo, compared with the free DOX. Moreover, in vitro studies showed high cytotoxic effects of C-P-CSOSA/DOX against MCF-7 (IC50 equal 1.45 ug/mL, where for free DOX IC50 was 5 times higher), and increased the generation of reactive oxygen spices with simultaneous activation of tumor apoptosis.

More recently, Jiang et al. reported delocalized lipophilic cations conjugated with synthesized anionic, cationic, and charge-neutral polymers [106] to improve mitochondrial targeting. Delocalized lipophilic cations conjugated anionic polymers accumulated in mitochondria when DLC-conjugated with cationic and charge-neutral polymers do not reach the target efficiently. Interestingly, side-chain modifications by hydrophobic hexyl or hydrophilic hydroxyl do not affect the mitochondrial localization, which was observed for 13 cell lines, e.g., adenocarcinoma human epithelial cell line A549, human cervical cancer cells HeLa or human umbilical vein endothelial cells HUVEC. Additionally, cyanine 3-tagged anionic polymers loaded with DOX demonstrated ability to inhibit the mitochondrial metabolic activity more effectively than free DOX after a 24 h treatment of HeLa cells.

### 3.4. Enzyme-Responsive Drug Delivery Systems

Alternatively, enzyme–sensitive conjugates can serve as a promising vehicle for cancer–targeting DDSs, capable of releasing the drug upon the hydrolysis of the amide bond of a specific peptide by proteases (Figure 6) [107]. Matrix metalloproteinases (MMPs) and cathepsin B (CB) are important representatives of proteases associated with cancer. MMPs are a family of zinc–dependent proteases involved in extracellular matrix degradation and tumor progression [108]. CB is a lysosomal cysteine protease, and its overexpression is correlated with invasion and metastasis of cancer cells [109]. Alternatively, DDSs can be activated by an enzyme to expose targeting ligands for cellular uptake.

Lee et al. synthesized dendrimer–methoxy poly(ethylene glycol)–DOX conjugate (Dendrimer–MPEG–DOX) using four amino acid (GFLG) peptides for CB–dependent targeting [110]. In vitro anti–tumor activity against CT26 colon carcinoma cells showed enhanced cytotoxicity of Dendrimer–MPEG–DOX. Importantly, Dendrimer–MPEG–DOX was more effective than DOX alone in inhibiting tumor growth in the mice CT26 tumor xenograft model. Additionally, it accumulated selectively in the tumor, whereas free DOX was equally distributed within the organism.

In 2020, Luo el al. developed DOX/nifuroxazide (NFX) co–loaded micelles (CLM) with enzyme–sensitive peptide GFLG. Hydroxypropyl methacrylate and oligo(ethylene glycol) methacrylate copolymer with GFLG peptide backbone was conjugated with DOX via acid–labile hydrazine bond [111]. Moreover, NFX was loaded via thin–film hydration and self–assembled into micelles. In vivo and ex vivo studies confirmed that CLM exerted anti–metastatic effect in orthotropic and lung metastatic breast cancer models. Along with the high anti–tumor efficacy of CLM, a reduced DOX cardiotoxicity was reported. On day 21 post–treatment, all mice treated with CLM (3 mg/kg of DOX and 5 mg/mL of NFX) survived with a tumor growth inhibition rate of 57%, whereas in the case of DOX–loaded micelles (3 mg/kg of DOX) inhibition rate was 27%.

Based on previous studies on the cleavage site specificity of MMP–2 and MMP–9, many MMPs-specific peptide sequences were proposed [112]. For example, Kratz et al. demonstrated that GPQRIAGQ peptide incorporated in DOX–human serum albumin conjugate was cleaved efficiently by activated MMP–2/9 [113]. Lee et al., in their study, employed two PEGylated peptide–DOX conjugate micelles using GPLGV and GPLGVRG peptides [114]. In vivo studies showed 72% (micelles using GPLGV) and 63.3% (micelles using GPLGVRG) tumor growth inhibition, compared with untreated control. In another study, two tumor activated prodrug–conjugated polystyrene nanoparticles (TAP–NPs), containing PLGSYL and GPLGIAGQN peptides, demonstrated substantial cytotoxicity toward HT1080, HDF, and HUVEC cells in a time-dependent manner [115]. More prominent effects were observed for HT1080 cells than for healthy and primary cells, and stronger inhibition was reported for TAP-NPs functionalized by GPLGIAGQN than by PLGSYL.

In 2012, Shi et al. synthesized cell-penetrating peptide–DOX conjugate (ACPP) with PLGLAG sensitive sequence that could release DOX in response to MMP–2 and MMP–9 [116]. The conjugate exerted high cytotoxic effects against HT–1080 cells which overexpress MMP–2/9, whereas only low cytotoxic activity was reported against MCF–7 cells characterized by low expression of MMPs. Upon addition of GM6001, an MMP inhibitor, the cellular uptake of ACPP by HT–1080 cells was reduced, demonstrating that the uptake is dependent on MMP activity.

A more investigative approach was used by Zhang et al., who designed DOX loaded on multifunctional envelope–type mesoporous silica nanoparticles (MEMSM) [117]. The surface of DOX–loaded MEMSM was functionalized with β–cyclodextrin (CD) via click chemistry through a disulfide bond. Next, mesoporous silica nanoparticles–CD was modified by the RGD peptide motif, a ligand for cell surface integrins, and subsequently by an MMP substrate PLGVR peptide, covalently coupled with polyanion (PASP) to form a protective layer. In vitro studies demonstrated efficient MEMSM uptake by the squamous carcinoma SCC–7 cells and human colon cancer HT–29 cells via RGD-mediated interactions following removal of PASP layer through cleavage of PLGVR by MMP–2, and DOX release in the presence of glutathione. Viability of both SCC–7 and HT–29 cells incubated with MEMSM (125 μg/mL) was reduced to 40%, and when MMP inhibitor was added, cell viability exceeded 70%, demonstrating enzyme–enhanced drug uptake and highlighting the role of MMPs in directing the drug to the tumor cells.

A similar approach with the application of another MMP substrate, KDPLGVC peptide, was proposed by Eskandari et al. [118]. The peptide was bound onto the surface of DOX–loaded MSN through amidation reaction, and then grafted with a gold nanoparticle–biotin conjugate (GNP) as end–capping and active targeting agent. Amount of DOX released from MSN–GNP–Bio@DOX increased to 82.5% in the presence of MMP–2 at pH 5.5, and due to the Au–S bond breaking, release decreased to 10% in the absence of MMP–2 and at pH 7.4. The DDSs demonstrated significant efficacy towards 4T1 biotin receptor–positive cancer cells overexpressing MMP–2 with a high level of cellular uptake and cell viability reduced to 4% after 72 h treatment. In contrast, viability of T47D breast cancer cells, which are characterized by a lack of biotin receptor and low MMPs expression, reached 60% upon the same treatment.

In other studies, DOX was conjugated to humanized IgG1 monoclonal antibody—trastuzumab by MMP–2 sensitive peptide linker (MAHNP–DOX) [119]. Trastuzumab targets human epidermal growth factor receptor 2 (HER2), and inhibits HER2–mediated malignant transformation [120]. In that study, 12–amino acid anti–HER2 peptide mimetic and GPLGLAGDD MMP–2 sensitive peptides were conjugated to DOX as active targeting strategy. MAHNP–DOX treatment decreased the growth rate of HER2 positive breast cancer cell lines BT474 and SKBR3 in a dose–dependent manner (IC_50_ values 747 and 110 nM for BT474 and SKBR3 cells, respectively). IC_50_ values were higher (1328.0 and 146.7 nM for BT474 and SKBR3 cells, respectively) when the cells had been pretreated with MMP–2 inhibitor. In vivo experiments on BT474 tumor–bearing mice showed that MAHNP–DOX resulted in 74.7% inhibition of tumor growth 25 days following the treatment. In mice treated with free DOX, inhibition of tumor growth was lower than in mice treated with MAHNP–DOX. Significant body weight loss was observed only in mice receiving free DOX rather than MAHNP–DOX.

Zhang et al. prepared dextran–coated Fe_3_O_4_ nanoparticles conjugated with DOX and chimeric monoclonal antibody cetuximab (DOX–NPs–Cet) for targeted anticancer therapy [121]. Dextran–coated Fe_3_O_4_ nanoparticles without DOX and Cet provided desirable stability and good biocompatibility, allowing for their application as drug carriers. DOX–NPs–Cet bound specifically to the epidermal growth factor receptor, which is overexpressed in non–small lung cancer A549 cells, and released DOX directly into the cells via endocytosis. Notably, DOX–NPs–Cet exhibited higher cytotoxicity against A549 cells than DOX–NPs (IC_50_ values after 48 h 0.22 µg/mL and 0.68 µg/mL, respectively).

In addition, transferrin receptor (TfR) overexpressed in many tumors seems to be a good target for selective drug delivery to enhance cellular uptake via TfR-mediated endocytosis [122]. In 2019, Li et al. designed TfR-targeted binding peptide analog BP9a (CAHLHNRS) coupled with DOX through *N*–succinimidyl–3–maleimidopropionate as a crosslinker [123]. Higher cytotoxic effects were observed toward HepG2 hepatoma cells overexpressing TfR than toward L-O2 normal human liver cells, whereas for free DOX, only poor selectivity for cancer cells was shown.

Moreover, some reports demonstrated that ferritin, an iron storage protein, successfully binds to TfR [124], and has been used to encapsulate chemotherapeutic drugs for targeted delivery. On the other hand, in the absence of iron, ferritin can form the hallow apoferritin, which has the same above-mentioned properties as ferritin.

Chen et al. used DOX-loaded apoferritin (DOX-APO) for delivering into the brain to inhibit the glioma tumor growth [125]. Here, the highly TfR-expressed C6 (glioma cell line) and bEnd.3 (mouse brain microvascular endothelial cells) cell lines were used to determine a significant cellular uptake via TfR receptor and efficient blood-brain barrier penetration by DOX-APO. In vivo studies using C6-beating mice demonstrated an accumulation of DOX-APO (1 mg/kg DOX) into brain tumor tissues with simultaneous longer mice survival time. Unfortunately, high liver accumulation was observed, which may introduce some limitation in the use of nanoparticle and required further analysis. Recently, H-chain modified apoferritin (TL-HFn) was used to deliver DOX into the cell nucleus after cellular uptake via TfR receptors and lysosome escape [126]. These studies proved that TL-HFn could be used as a safe carrier for small molecules without any cytotoxic effects against HeLa cells. After DOX encapsulation into TL-HFn, the cytotoxicity was observed in a wide range of concentrations (0.016–4.00 mg/mL) and was comparable to the action of free DOX.

Developing a carrier that induces apoptosis specifically in tumors using tumor necrosis factor–related apoptosis–inducing ligand (TRAIL) represents another exciting DDSs approach [127]. Jiang et al. developed DOX encapsulated liposomes with TRAIL and cell–penetrating peptide R8H3, further coated by hyaluronic acid–cross–linked gel shell (TRAIL/DOX-Gelipo). Hyaluronidase, an extracellular enzyme overexpressed in tumors, degraded hyaluronic acid–cross–linked gel shell, exposed R8H3 to facilitate the cellular uptake via endocytosis, and released TRAIL [128]. After the endosomal escape, DOX accumulated into the cell nucleus to trigger apoptosis. TRAIL/DOX–Gelipo treated by hyaluronidase showed cytotoxicity toward MDA MB 231 cells significantly higher than DOX–Gelipo without TRAIL, with IC_50_ value 83 ng/mL (vs. 569 ng/mL). Additionally, TRAIL/DOX–Gelipo triggered high DOX accumulation in tumor and efficient tumor growth suppression.

## 4. Conclusions

In this review, we discussed DOX delivery systems and their evolution in the last few years. Since Doxil^®^ and Myocet^®^, many different DDS concepts appeared to overcome biological barriers and reduce drug side effects. All summarized technologies share common ideas of efficient pharmaceutical cargo transportation through the whole body, followed by DOX maximized accumulation in cancer tissue, improved through controlled release into cancer cells by a wide spectrum of stimuli. For example, DDSs can be sensitive to chemical and physical stimuli such as pH changes or light, as well as biological ones, e.g., enzymes overexpressed by cancer cells. Therefore, choosing the type of delivery system and its design is critical. For these reasons, new synthetic approaches and polymerization methods to create DDSs in a controlled manner with desired features in a relatively short time are a subject of intensive studies. A tremendous amount of effort is being put into maximized execution of DOX therapeutic effects towards targeted cells. Future benefits, that are being expected to be brought with engineered nanotechnology in DDs, involve overcoming possible physiological conditions against DOX on its road to the targeted site, simultaneously providing sufficient concentration of the drug to cancerous cells in a specified therapeutic window. Carefully designed nanocarriers would also harness their potential to synergistically support DOX in decreasing tumor developments, accompanied by reduced systemic harmfulness. For all the researchers, it is also crucial to consider drawbacks that potentially can be faced in the future during technology translation from the laboratory bench to the clinical trials and product administration to patients. There is no doubt that the DDSs described in this review demonstrate the potential to form efficient and targeted systems for future innovations in the field of DOX delivery. However, many challenges must be improved to achieve clinical trials and FDA approval. In our opinion, biosafety and biocompatibility are one of the most important parameters of DDSs, and their lack of toxicity may reduce the risk of side effects and enhance therapeutic outcomes. As for polymeric DDSs, the major obstacle is their high complexity and architecture which required advanced polymerization methods to obtain a polymer with high efficiency and without impurities. Despite their hurdle and difficulties with synthesis, some simple polymers, like PEG, are commonly used to increase the solubility and biocompatibility of DDSs. Moreover, the rational design of DDSs might be improved by stimuli-responsive moieties conjugated to previously synthesized polymers. Hundreds of stimuli-responsive DDSs have been reported up to now, and showed many advantages, like improvement of pharmacokinetics and accumulation of DOX in the tumor site. Besides, they also may decrease off-target effects and metastasis. However, the application of stimuli-responsive DDSs requires better control of drug dose which is released from the carrier in a time-dependent manner. Unfortunately, many of them are not suitable for in vivo studies, because of non-biodegradable character or lack of high therapeutic efficacy. On the other hand, targeted DDSs using, e.g., receptors, are capable to overcome biological barriers associated with cellular uptake by receptor/ligand-mediated endocytosis. Great efforts have been made based on binding ligands and open new opportunities for cellular targeting and DDSs selectivity. This approach is related to surface binding by DDSs and further mechanism of drug release into the cytosol after the endosomal escape. Targeted DDSs have shown promise as potential therapeutic agents, but a detailed understating of their mechanism of action is needed to avoid nonspecific interactions and achieve delivery to different cancer cells. Given this, DDSs still have a long way to go in terms of optimization and innovation in design and development. We believe that thoroughly reviewed information and critical evaluation of the work progress on DDS in recent years would inspire the creation of new strategies for the DOX ideal carrier development.

## Figures and Tables

**Figure 1 materials-14-02135-f001:**
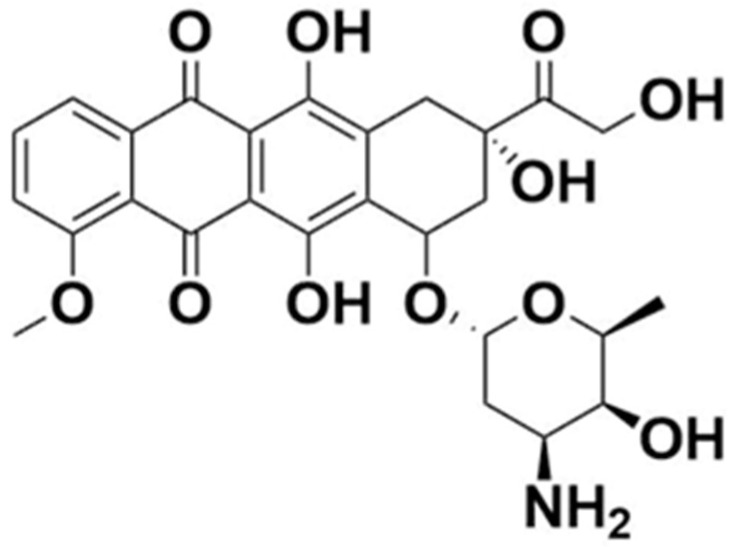
Chemical structure of doxorubicin.

**Figure 2 materials-14-02135-f002:**
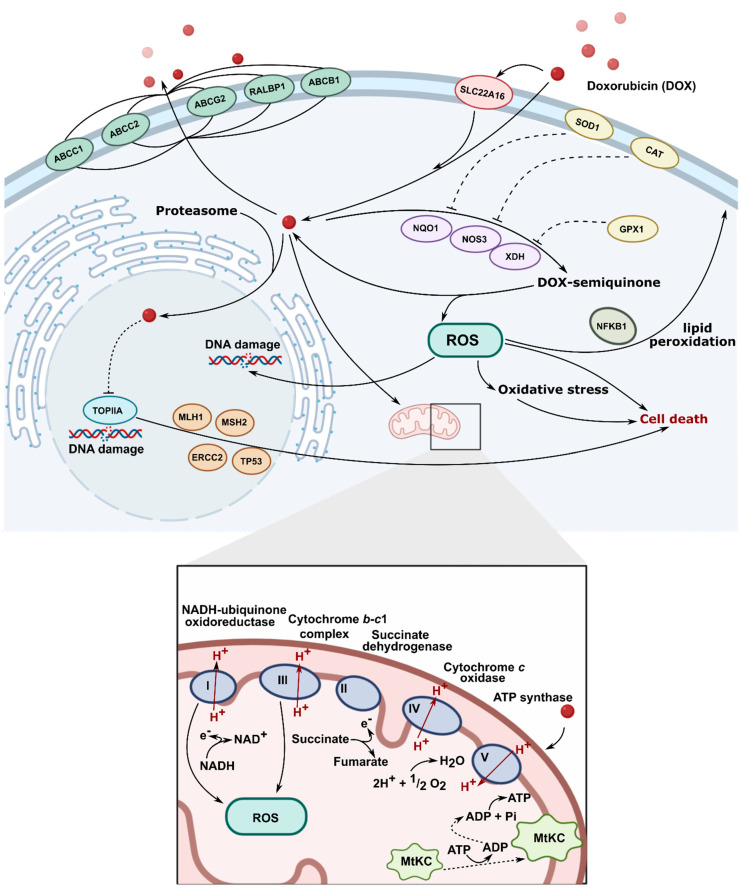
Molecular mechanism of action of doxorubicin (TOPIIA-topoisomerase II, ROS-reactive oxygen species).

**Figure 3 materials-14-02135-f003:**
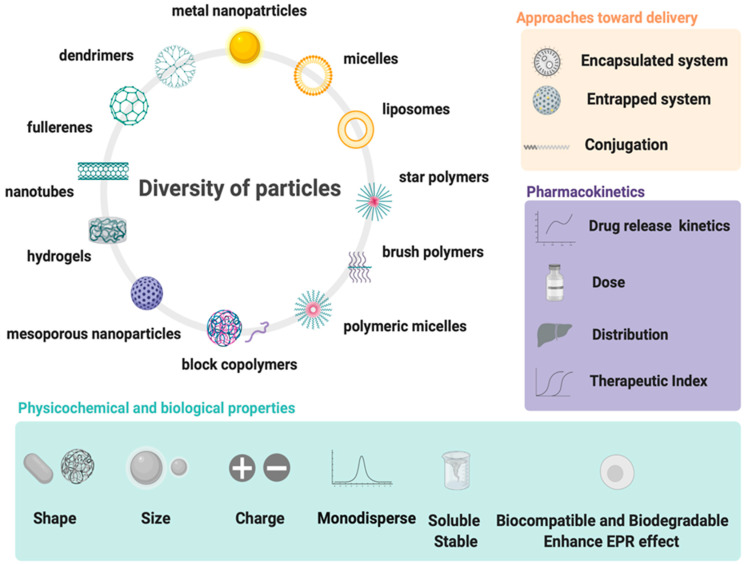
Design and properties requirements for drug delivery systems using wide spectrum of particles.

**Figure 4 materials-14-02135-f004:**
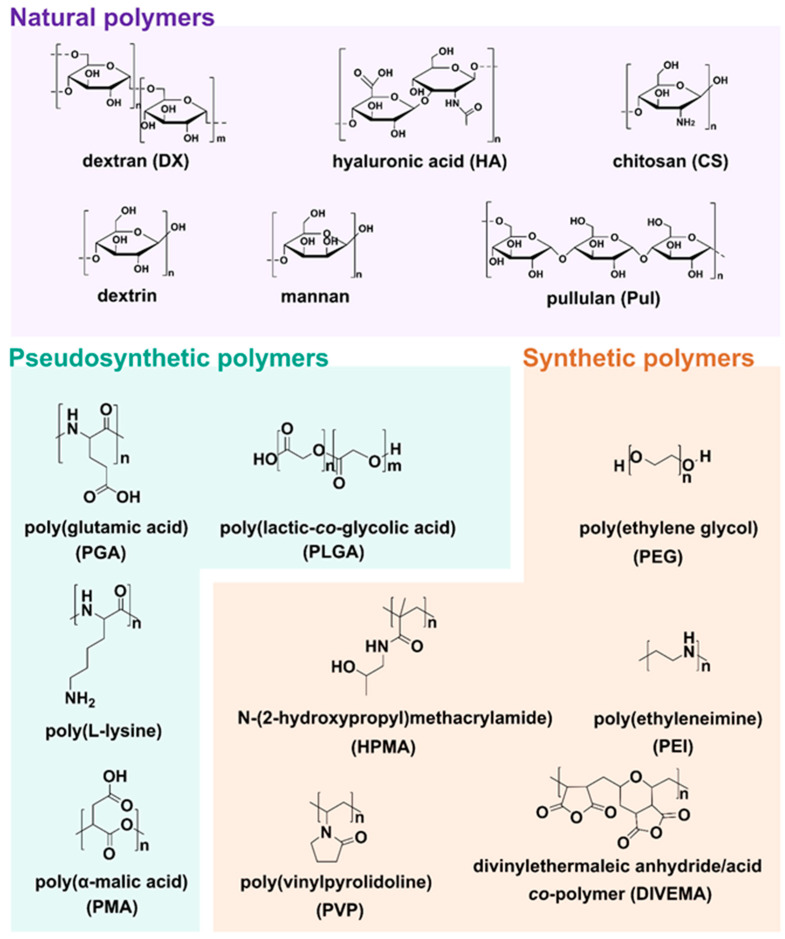
Structures of natural, pseudosynthetic, and synthetic polymers used for drug delivery applications.

**Figure 5 materials-14-02135-f005:**
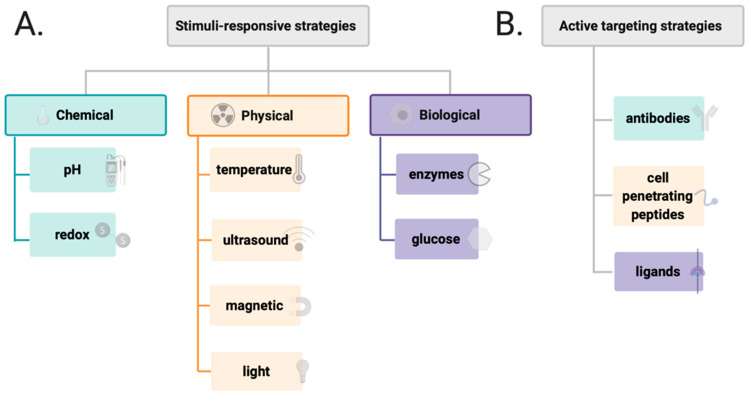
Different types of stimulus appleid in the design of drug delivery system (**A**) and various factors affecting active targenting of drug delivery systems (**B**).

**Figure 6 materials-14-02135-f006:**
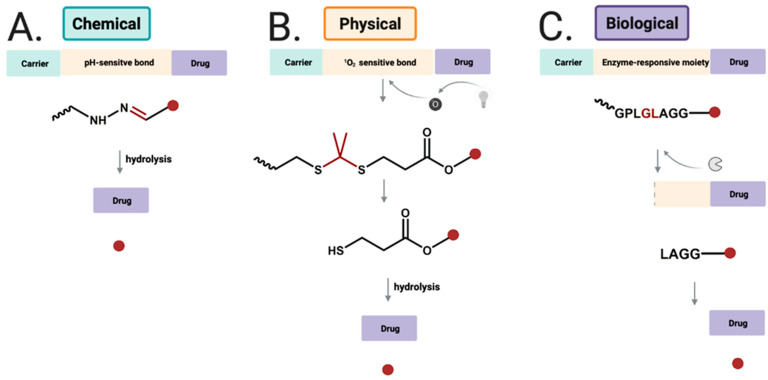
Drug release strategies for (**A**) chemical-responsiveDDSs by hydolysis of hydrazone bond to release free drug; (**B**) physical-responsive DDSs using near infrared light to generate singlet oxygen, which undergoes reaction with thioketal group; (**C**) biological-sensitive DDSs cleaved byenzyme, and further cleaved to release free drug.

## Data Availability

Data sharing is not applicable to this article.

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
