# Peer review of "Polymeric Nanocarriers: A Transformation in Doxorubicin Therapies"

_materials, 2021, doi:10.3390/ma14092135_

Round 1

Reviewer 1 Report

The manuscript is focused on the clinically important topic of drug delivery systems with doxorubicin. However, the report is only a summary of facts published so far without any attempting to analyze it and without any future prospects. In addition, a number of clinically relevant facts are missing (indications for combinations containing doxorubicin in various tumors, the main side effect of doxorubicin is myelotoxicity, more detailed information on registered DDSs with doxorubicin). Nephrotoxicity caused by anthracyclines is relevant in mice, but does not occur in clinical practice nor is reported in the medical literature. The authors state: “we summarized the progress achieved within the last few years in the field of co–delivery for DOX and siRNA to overcome multidrug resistance” and there is no mention of co-delivery of DOX and siRNA, although a number of studies have been published on this topic, eg. Tieu T et al., Nanobody-displaying porous silicon nanoparticles for the co-delivery of siRNA and doxorubicin. Biomater Sci. 2021;9:133; Ashrafizadeh M et al. Progress in Natural Compounds/siRNA Co-delivery Employing Nanovehicles for Cancer Therapy. ACS Comb Sci. 2020;22:669; Zhang X et al. Charge reversible hyaluronic acid-modified dendrimer-based nanoparticles for siMDR-1 and doxorubicin co-delivery. Eur J Pharm Biopharm. 2020;154:43 (I am not the co-author of any of them). I miss the mention of apoferritin as a nanocarrier, although authors described the TfR-targeted binding peptide analog, apoferritin also binds to TfR. The EPR effect is not explained, Figure 3, to which reference is made in the text, also does not explain it.

For these reasons, I do not recommend accepting the manuscript.

Author Response

We would like to thank the Referee for his/her additional comments which helped us to improve the quality of the manuscript. Our detailed response and the list of corrections addressing all the remarks are listed below:

Referee #1

However, the report is only a summary of facts published so far without any attempting to analyze it and without any future prospects.

Authors’ response: In order to improve the future prospects, we introduced the following change in the manuscript (see line 520-544):

“In this review, we discussed DOX delivery systems and their evolution in the last few years. Since Doxil® and Myocet®, many different DDSs concepts appeared to overcome biological barriers and reduce side effects. The delivery of DOX and its maximized accumulation in cancer tissue could be improved through controlled release into cancer cells by a wide spectrum of stimuli. For example, DDSs can be sensitive to chemical and physical stimuli such as pH changes or light, as well as biological ones, e.g. enzymes overexpressed by cancer cells. Therefore, choosing the type of delivery system and its design is critical. For these reasons, new synthetic approaches and polymerization methods to create DDSs in a controlled manner with desired features in a relatively short time are a subject of intensive studies. There is no doubt that the DDSs mentioned in this review demonstrate the potential to form efficient and targeted systems for future innovations in the field of DOX delivery, but there are still many ways to generate the ideal carrier.”

was substituted with

“In this review, we discussed DOX delivery systems and their evolution in the last few years. Since Doxil® and Myocet®, many different DDSs concepts appeared to overcome biological barriers and reduce drug side effects. All summarized technologies share common ideas of efficient pharmaceutical cargo transportation through the whole body, followed by DOX maximized accumulation in cancer tissue, improved through controlled release into cancer cells by a wide spectrum of stimuli. For example, DDSs can be sensitive to chemical and physical stimuli such as pH changes or light, as well as biological ones, e.g. enzymes overexpressed by cancer cells. Therefore, choosing the type of delivery system and its design is critical. For these reasons, new synthetic approaches and polymerization methods to create DDSs in a controlled manner with desired features in a relatively short time are a subject of intensive studies. A tremendous amount of effort is being put into maximized execution of DOX therapeutic effects towards targeted cells. Future benefits, that are being expected to be brought with engineered nanotechnology in DDs, involve overcoming possible physiological conditions against DOX on its road to the targeted site, simultaneously providing sufficient concentration of the drug to cancerous cells in a specified therapeutic window. Carefully designed nanocarriers would also harness their potential to synergistically support DOX in decreasing tumor developments, accompanied by reduced systemic harmfulness. For all the researchers, it is also crucial to consider drawbacks that potentially can be faced in the future during technology translation from the laboratory bench to the clinical trials and product administration to patients. There is no doubt that the DDSs described in this review demonstrate the potential to form efficient and targeted systems for future innovations in the field of DOX delivery. We believe that thanks to thoroughly reviewed information and critical evaluation of the work progress on DDS in recent years would inspire the creation of new strategies for the DOX ideal carrier development.”

In addition, a number of clinically relevant facts are missing (indications for combinations containing doxorubicin in various tumors, the main side effect of doxorubicin is myelotoxicity, more detailed information on registered DDSs with doxorubicin).

Authors’ response: The subsequent sentence has been added to the manuscript: „ Apart from its high efficacy in monotherapy (especially in treatment of metastatic breast cancer), several combination therapies including DOX were also developed. The combination of DOX with cyclophosphamide, vincristine, and prednisone is used for treatment of diffuse large cell non-Hodgkin's lymphomas (Abraham, R., et al. (1996). A Risk-Benefit Assessment of Anthracycline Antibiotics in Antineoplastic Therapy. Drug Safety, 15(6), 406–429). The combination of DOX with bleomycin, vincristine and dacarbazine is beneficial and well tolerated in patients with Hodgkin's lymphoma (Abraham, R., et al. (1996). A Risk-Benefit Assessment of Anthracycline Antibiotics in Antineoplastic Therapy. Drug Safety, 15(6), 406–429). Several combination regimens consisting of DOX are used for treatment of breast cancer (DOX with cyclophosphamide and/or taxotere, DOX with cyclophosphamide and fluorouracil” (see lines 41-48).

The authors state: “we summarized the progress achieved within the last few years in the field of co–delivery for DOX and siRNA to overcome multidrug resistance” and there is no mention of co-delivery of DOX and siRNA, although a number of studies have been published on this topic, eg. Tieu T et al., Nanobody-displaying porous silicon nanoparticles for the co-delivery of siRNA and doxorubicin. Biomater Sci. 2021;9:133; Ashrafizadeh M et al. Progress in Natural Compounds/siRNA Co-delivery Employing Nanovehicles for Cancer Therapy. ACS Comb Sci. 2020;22:669; Zhang X et al. Charge reversible hyaluronic acid-modified dendrimer-based nanoparticles for siMDR-1 and doxorubicin co-delivery. Eur J Pharm Biopharm. 2020;154:43 (I am not the co-author of any of them).

Authors’ response: The subsequent sentence has been removed from the manuscript.

I miss the mention of apoferritin as a nanocarrier, although authors described the TfR-targeted binding peptide analog, apoferritin also binds to TfR. The EPR effect is not explained, Figure 3, to which reference is made in the text, also does not explain it.

Authors’ response: The subsequent sentence has been added to the manuscript: Moreover, some reports demonstrated that ferritin, an iron storage protein, successfully binds to TfR (Liang, M., et al., (2014) H-ferritin-nanocaged doxorubicin nanoparticles specifically target and kill tumors with a single-dose injection, PNAS, 111(41), 14900-14905), and has been used to encapsulate chemotherapeutic drugs for targeted delivery. On the other hand, in the absence of iron, ferritin can form the hallow apoferritin, which has the same above-mentioned properties as ferritin. Chen et. al used DOX-loaded apoferritin (DOX-APO) for delivering into the brain to inhibit the glioma tumor growth (Chen, Z., el al., (2017) Apoferritin nanocage for brain targeted doxorubicin delivery, Molecular pharmaceutics, 14(9), 3087-3097). Here, the highly TfR-expressed C6 (glioma cell line) and bEnd.3 (mouse brain microvascular endothelial cells) cell lines were used to determine a significant cellular uptake via TfR receptor and efficient blood-brain barrier penetration by DOX-APO. In vivo studies using C6-beating mice demonstrated an accumulation of DOX-APO (1 mg/kg DOX) into brain tumor tissues with simultaneous longer mice survival time. Unfortunately, high liver accumulation was observed, what may introduce some limitation in the use of nanoparticle and required further analysis.  Recently, H-chain modified apoferritin (TL-HFn) was used to deliver DOX into the cell nucleus after cellular uptake via TfR receptors and lysosome escape (Pan,X. Et al., (2021) Tetralysine modified H-chain apofkrritin mediated nucleus delivery of chemotherapy drugs synchronized with passive diffusion, Journal of Drug Delivery Science and Technology, 61, 102132). These studies proved that TL-HFn could be used as a safe carrier for small molecules without any cytotoxic effects against HeLa cells. After DOX encapsulation into TL-HFn, the cytotoxicity was observed in a wide range of concentrations (0.016 - 4.00 mg/mL) and was comparable to the action of free DOX.” (See lines 488-505).

Reviewer 2 Report

The manuscript reviews recent developments in drug delivery systems, more specifically polymer-based nanocarriers targeting doxorubicin therapies. The subject is both relevant and current, and a lot of attention has been paid on it. Being aware of the existence of numerous published review articles on the subject, the authors propose an update targeted at the title materials and underlying function, with a special focus on works carried out over the last decade. Although this is not an original work in every respect, the result is an up-to-date compilation with potential interest for readers of Materials. The acceptable content is supported by a careful form (the reviewer detected no typos but recommends that "Molecular dynamic" be replaced by "Molecular dynamics" in line 222). The manuscript is publishable in its current form. 

Author Response

We would like to thank the Referee for his/her additional comments which helped us to improve the quality of the manuscript. Our detailed response and the list of corrections addressing all the remarks are listed below:

Referee #2

The acceptable content is supported by a careful form (the reviewer detected no typos but recommends that "Molecular dynamic" be replaced by "Molecular dynamics" in line 222). 

Authors’ response: According to the Reviewer's suggestion, „Molecular dynamic” has been replaced by „Molecular dynamics” (see line 235).

Reviewer 3 Report

Piosik and coworkers summarized the delivery of doxorubicin using polymeric materials in the past decade. It is an important aspect of polymeric delivery vehicles in the delivery of small molecules. 

  1. The authors did mention the (cardio)toxicity of doxorubicin in the manuscript. It is highly recommended to extract these information in a separate section (can be short). Readers do need to be aware of the pros and cons of this drug by easily indexing the review.
  2. The authors overall did a good job in the figures. Figure 6 demonstrates a clear example of how enzymatic responsive drug delivery systems work. Such figures shall be extended to other modes of actions in Figure 5. 
  3. Another missing figure is the representative chemical/polymeric structures for the categories listed in Figure 5. It can be either supplemented by a figure or a table. 
  4. One possible mechanism of action for doxorubicin is the disruption on mitochondrial activities, which is also mentioned by the authors in the review. However, examples on mitochondrial targeting delivery of doxorubicin using polymeric materials are missing. Several references shall be summarized in the revised version: DOI: 10.1039/C8PY01504J; DOI: 10.1021/acs.bioconjchem.0c00079; DOI: 10.1038/s41598-020-65450-x. 

Author Response

We would like to thank the Referee for his/her additional comments which helped us to improve the quality of the manuscript. Our detailed response and the list of corrections addressing all the remarks are listed below:

Referee #3

The authors did mention the (cardio)toxicity of doxorubicin in the manuscript. It is highly recommended to extract these information in a separate section (can be short). Readers do need to be aware of the pros and cons of this drug by easily indexing the review.

Authors’ response:  The subsequent sentence has been added to the manuscript: „ DOX induces myelosuppression, mainly in the form of leukopenia (principally granulocytopenia), neutropenia, or thrombocytopenia, with up to 80% of patients treated with conventional doses of DOX being affected (Abraham, R., et al. (1996). A Risk-Benefit Assessment of Anthracycline Antibiotics in Antineoplastic Therapy. Drug Safety, 15(6), 406–429; Julka, P. K., et al. (2008). A phase II study of sequential neoadjuvant gemcitabine plus doxorubicin followed by gemcitabine plus cisplatin in patients with operable breast cancer: Prediction of response using molecular profiling. British Journal of Cancer, 98(8), 1327–1335). The severity of myelotoxicity is dose-dependent, therefore it represents the major dose-limiting side effect of anthracycline therapy.

Besides the heart and bone marrow, toxic effects of DOX are observed also in the liver and brain [25]. Other side effects of DOX include nausea and vomiting, stomatitis, mucositis, alopecia, and neurologic disturbances (dizziness, hallucinations) (Liu, J., et al. (2006). Quality of life analyses in a clinical trial of DPPE (tesmilifene) plus doxorubicin versus doxorubicin in patients with advanced or metastatic breast cancer: NCIC CTG Trial MA.19. Breast Cancer Research and Treatment, 100(3), 263–271). Severe vesicant reactions might also occur upon extravasation of DOX which can lead to severe local tissue necrosis and reduced mobility in the adjacent joints.” (See lines 104-113).

The authors overall did a good job in the figures. Figure 6 demonstrates a clear example of how enzymatic responsive drug delivery systems work. Such figures shall be extended to other modes of actions in Figure 5. 

Authors’ response: Figure 6 has been extended to chemical and physical models of actions. Additionally, structures of chemical bonds/peptide linkers and their mechanism for releasing the desired drug have been added.

Another missing figure is the representative chemical/polymeric structures for the categories listed in Figure 5. It can be either supplemented by a figure or a table. 

Authors’ response: We demonstrated many structures of natural, pseudosynthetic and synthetic polymers used for drug delivery application in Figure 4, which have been used in different combinations for stimuli-sensitive drug delivery systems and were described in the main text of manuscript. No further action has been undertaken.  

One possible mechanism of action for doxorubicin is the disruption on mitochondrial activities, which is also mentioned by the authors in the review. However, examples on mitochondrial targeting delivery of doxorubicin using polymeric materials are missing. Several references shall be summarized in the revised version: DOI: 10.1039/C8PY01504J; DOI: 10.1021/acs.bioconjchem.0c00079; DOI: 10.1038/s41598-020-65450-x. 

Authors’ response: The subsequent sentence has been added to the manuscript: „ Interestingly, despite many unique characteristics of cancer cells, like low extracellular pH and hypoxia, their hyperpolarized mitochondria opening new directions to targeted drug delivery (Indran, I.R. et al., (2011) Recent advances in apoptosis, mitochondria and drug resistance in cancer cells, Biochimica et Biophysica Acta (BBA)- Bioenergetics, 1807(6), 735-745). Many reports demonstrated potential applications of modified polymers to locate drugs inside the mitochondria (Biswas, S., et al., (2012) Liposomes loaded with paclitaxel and modified with novel triphenylphosphonium-PEG-PE conjugate possess low toxicity, target mitochondria and demonstrate enhanced antitumor effects in vitro and in vivo, Journal of Controlled Release, 159(3), 393-402, Biswas, S., et al., (2011) Surface modification of liposomes with rhodamine-123-conjugated polymer results in enhanced mitochondrial targeting, Journal of Drug Targeting, 19(7), 552-561). In 2019, Tan et al. presented micelles for DOX delivering, using glycolipid polymer chitosan-stearic acid (CSOSA), which was modified by lipophilic (4-carboxybutyl)triphenylphosphonium bromide (CTPP) cations, to form mitochondria-targeted DDSs (C-P-CSOSA/DOX) (Tan, Y., et al., (2019) In vivo programming of tumor mitochondria-specific doxorubicin delivery by a cationic glycolipid polymer fo enhanced anti tumor activity, Polymer chemistry, 10(4), 512-525). The relatively small C-P-CSOSA/DOX particles, with a size around 100 nm, showed higher cellular uptake in human breast adenocarcinoma cells (MCF-7 cell line) than in human normal liver cells (L02 cell line). Importantly, C-P-CSOSA/DOX demonstrated efficient co-localization into mitochondria in vitro and in vivo, compared with the free DOX. Moreover, in vitro studies showed high cytotoxic effects of C-P-CSOSA/DOX against MCF-7 (IC50 equal 1.45 ug/ml, where for free DOX IC50 was 5 times higher), and increased the generation of reactive oxygen spices with simultaneous activation of tumor apoptosis. More recently, Jiang et al. reported delocalized lipophilic cations conjugated with synthesized anionic, cationic, and charge-neutral polymers (Jiang, Z., el al., (2020) Anionic polymers promote mitochondrial targeting of delocalized lipophilic cations, Bioconjugate Chemistry, 31(5), 1344-1353) to improve mitochondrial targeting. Delocalized lipophilic cations conjugated anionic polymers accumulated in mitochondria when DLC-conjugated with cationic and charge-neutral polymers do not reach the target efficiently. Interestingly, side-chain modifications by hydrophobic hexyl or hydrophilic hydroxyl do not affect the mitochondrial localization, which was observed for 13 cell lines, e.g. adenocarcinoma human epithelial cell line A549, human cervical cancer cells HeLa or human umbilical vein endothelial cells HUVEC Additionally, cyanine 3-tagged anionic polymers loaded with DOX demonstrated ability to inhibit the mitochondrial metabolic activity more effective than free DOX after 24 h treatment of HeLa cells.” (See lines: 364-388).

Round 2

Reviewer 1 Report

The revision slightly improved the level of the manuscript, although I lack an evaluation of the advantages and disadvantages of individual DDSs. The sentence "There is no doubt that the DDSs described in this review demonstrate the potential to form efficient and targeted systems for future innovations in the field of DOX delivery" should be changed to "some of the DDSs ..." and it would be appropriate to add an opinion of authors which of them and why seems to be promising.

Author Response

Thank you very much for your comments. We added an appropriate fragment of text in Conclusions - marked red.

Reviewer 3 Report

The authors have properly addressed the comments from previous reviewers. 

Author Response

Thank you very much for your comments.